# Association between Sarcopenia and Reduced Bone Mass: Is Osteosarcopenic Obesity a New Phenotype to Consider in Weight Management Settings?

**DOI:** 10.3390/life14010021

**Published:** 2023-12-21

**Authors:** Antonino De Lorenzo, Leila Itani, Paola Gualtieri, Massimo Pellegrini, Marwan El Ghoch, Laura Di Renzo

**Affiliations:** 1Section of Clinical Nutrition and Nutrigenomic, Department of Biomedicine and Prevention, University of Tor Vergata, Via Montpellier 1, 00133 Rome, Italy; delorenzo@uniroma2.it (A.D.L.); paola.gualtieri@uniroma2.it (P.G.); laura.di.renzo@uniroma2.it (L.D.R.); 2Department of Nutrition and Dietetics, Faculty of Health Sciences, Beirut Arab University, Riad El Solh, Beirut 11072809, Lebanon; l.itani@bau.edu.lb; 3Center for the Study of Metabolism, Body Composition and Lifestyle, Department of Biomedical, Metabolic and Neural Sciences, University of Modena and Reggio Emilia, Via Campi 278, 41125 Modena, Italy; massimop@unimore.it

**Keywords:** bone mass, DXA, muscle mass, obesity, sarcopenic obesity

## Abstract

Sarcopenic obesity (SO) is a frequent phenotype in people with obesity; however, it is unclear whether this links with an impaired bone status. In this study, we aimed to investigate the association between SO and low bone mass, and to assess the prevalence of a new entity that combines excessive fat deposition, reduced muscle mass and strength, and low bone mass defined as osteosarcopenic obesity (OSO). Body composition was completed by a DXA scan in 2604 participants with obesity that were categorized as with or without SO, and with low or normal bone mineral content (BMC). Participants with both SO and low BMC were defined as OSO. Among the entire sample, 901 (34.6%) participants met the criteria for SO. This group showed a reduced mean BMC (2.56 ± 0.46 vs. 2.85 ± 0.57, *p* < 0.01) and displayed a higher prevalence of individuals with low BMC with respect to those without SO (47.3% vs. 25.9%, *p* < 0.01). Logistic regression analysis showed that the presence of SO increases the odds of having low BMC by 92% [OR = 1.92; 95% CI: (1.60–2.31), *p* < 0.05] after adjusting for age, body weight, and body fat percentage. Finally, 426 (16.4%) out of the total sample were affected by OSO. Our findings revealed a strong association between SO and reduced bone mass in adults with obesity, and this introduces a new phenotype that combines body fat, muscle, and bone (i.e., OSO) and appears to affect 16% of this population.

## 1. Introduction

Obesity is a condition defined as excessive fat deposition in the adipose tissue, and commonly its classification relies on body mass index (BMI) cut-off points in adults [1]. Specifically, a BMI ≥ 30 kg/m^2^ is suggestive of obesity, and this cut-off point is valid for all age groups and genders in the Caucasian population [1], if not otherwise specified in certain populations [2,3]. In community and clinical settings, obesity is one of the most significant health problems worldwide [4], since it is associated with a major risk factor for several medical (i.e., type 2 diabetes, cardiovascular diseases, dyslipidemia, sleep apnea, osteoarthritis, and others) [5,6] and psychosocial morbidities (i.e., depression, eating disorders, and impaired health-related quality of life (HRQoL) [7,8,9,10]), as well as disability [11] and increased rates of mortality [12]. The recent World Health Organization (WHO) report on obesity prevalence in Europe was concerning, since they stated that more than 50% of citizens in this area are either affected by obesity or are overweight [13]. This has prompted international scientific associations and societies dealing with obesity (i.e., European Association for the Study of Obesity) to raise their voices through the establishment of guidelines that recommend the early screening of this condition and push for a wide range of weight loss interventions possibly in the early stages [14,15,16].

In this context, a phenotype of obesity known as sarcopenic obesity (SO), characterized by increased fat accumulation and reduced muscle mass and strength, was noticed to be prevalent in several clinical settings, especially those for weight management [17], with wide ranges of reported prevalence [18]. Moreover, individuals with SO during weight loss programs were associated with increased cardio-metabolic risk factors and higher odds of having dyslipidemia, hypertension, type 2 diabetes, and impaired health-related quality of life (HRQoL), with respect to their relative counterparts who were obese but without SO [19]. Not only this, but they appeared to have also poorer weight loss outcomes, in particular, higher rates of attrition and early drop out (i.e., interruption of treatment), and more difficulties with weight loss maintenance in the longer term [19]. For these reasons, researchers pronounced that this population (i.e., SO) demand specific attention, especially in weight management settings [20].

Among this particular population (i.e., SO), certain research areas and topics have not been sufficiently investigated and therefore are still not fully understood, such as the link between SO and bone health in treatment-seeking adults with obesity. Despite this fact, prematurely, some researchers have recently described a very new phenotype that combined, in addition to the increased fat deposition and reduced muscle mass and strength seen in SO, another component of low bone mass [21], denominating it as osteosarcopenic obesity (OSO) [22,23,24]. However, a lot of debate surrounds this new phenotype, to the extent that some researchers argue about its existence, whereas others support the opposite [25,26]. To the best of our knowledge, little is known about the topic due to the paucity of studies, especially in weight management settings. For this reason, this population still requires more investigation and better understanding.

Based on all of these considerations, the current study aims firstly to detect the potential association between the presence of SO and the risk of having low bone mass, and secondly, to assess the prevalence of a new phenotype, the so-called OSO, in a nutritional setting in adults with obesity.

## 2. Materials and Methods

### 2.1. Participants and Design of the Study

This study is a retrospective study. Participants were consecutive and voluntarily recruited in the Division of Clinical Nutrition at the Department of Biomedicine and Prevention, University of Rome “Tor Vergata”, Italy, between June 2018 and May 2022. The patients were included in this study if they had an age ≥18 years, had obesity (BMI ≥ 30 kg/m^2^), and were required to have a DXA scan. Patients were categorized as with or without SO and were divided into a low, middle, or high tertile according to the sample distribution of bone mineral content (BMC), and based on that, they were classified as ‘Low-BMC’ or ‘Normal-BMC’. Patients were excluded if they were aged <18 years, were taking prescribed medications that affect body composition, or had any medical condition associated with weight loss or severe psychiatric disorders. Accordingly, 2604 participants representing both genders were included in this study. This study was conducted in accordance with the Declaration of Helsinki and informed written consent was obtained.

### 2.2. Body Weight and Height

Body weight and height were measured using an electronic weighing scale (SECA 2730-ASTRA, Hamburg, Germany) and a stadiometer, with individuals wearing light clothes and no shoes. The BMI was then calculated according to the standard formula as follows [27]:Body weight (kg) ÷ height^2^ (m).

### 2.3. Body Composition

Body composition was determined using a DXA (DXA, GE Medical Systems, Chicago, IL, USA) fan beam scanner, which measures both whole and regional compartments in terms of fat, lean mass, and bone mass, according to standardized instructions given to patients regarding the testing procedure, as described elsewhere [28].

SO was defined according to Batsis et al.’s criterion, which is the ratio of appendicular skeletal muscle mass (ASM) (kg) adjusted for body mass index (kg/m^2^) (ASM/BMI), with cut-off points <0.789 kg/m^2^ for males and <0.512 kg/m^2^ for females, and patients with ASM/BMI below these cut-off points were categorized as individuals with SO [29]. Low BMC was defined as those who fell into the lowest BMC for height tertile, otherwise they were classified as normal BMC if they fell into the medium or highest tertiles [30]. Finally, OSO was defined as a new entity that combines SO and low BMC [25].

### 2.4. Statistical Analysis

Descriptive statistics are presented as means and standard deviations for continuous variables. Frequencies and proportions were used to present categorical variables’ distribution. Student’s *t*-test and chi-squared test for independence were used for mean comparison and to test the difference in the categorical variables’ distribution, respectively. Logistic regression models were used to calculate the odds of low BMC given the presence of SO. For this purpose, the BMC to height ratio was calculated (kg/m) as a linear variable and categorized into sex-specific tertiles. BMC to height tertiles were determined separately for males and females, and an ordinal variable was created, classifying people in the first, second, or third tertile categorized based on sex-specific distribution. The ordinal variable was regrouped into a binary variable to facilitate the use of a binary logistic regression model, with the outcome of interest being in the first tertile of “BMC to height” ratio, as an indicator of low BMC. The logistic regression model was not adjusted for sex and height because “BMC to height” sex-specific tertiles were used. Potential confounders included were weight and body fat percentage (BF%). All statistical analyses were performed using Statistical Package for Social Sciences 25 (SPSS citation: IBM Corp. Released 2017. IBM SPSS Statistics for Windows, Version 25.0. Armonk, NY, USA: IBM Corp), with tests considered significant at *p* < 0.05.

## 3. Results

A total of 2604 participants were included in the study, with a mean age of 48.44 ± 14.31 years, mean BMI of 33.87 ± 2.73 kg/m^2^, and 57.3% females (*n* = 1429) (Table 1). Those with SO (901 (34.6%)) were characterized by being significantly older (53.48 ± 14.38 vs. 45.77 ± 13.53 years), mostly males (51.6% vs. 48.4%) with a lower mean weight (90.41 ± 13.59 kg vs. 94.41 ± 12.08 kg), and having a higher BMI (34.52 ± 2.82 vs. 33.53 ± 2.62 kg/m^2^), BF (40.61 ± 7.40 vs. 39.02 ± 7.37 kg), and BF% (45.18 ± 6.72 vs. 41.65 ± 7.31%) when compared to those without SO (Table 1).

Moreover, individuals with SO displayed inferior mean BMC (2.56 ± 0.46 vs. 2.87 ± 0.57 kg) and BMC/h (1.58 ± 0.23 vs. 1.69 ± 0.27 kg/m) when compared to those without SO (Table 1) (Figure 1). The SO group included more individuals with low BMC (47.3% vs. 25.9%) than those in the group without SO (Figure 2). In line with that, the logistic regression analysis revealed almost double the risk of low BMC [OR = 2.57; 95% CI: (2.17–3.04), *p* < 0.05] in the presence of SO (Table 2). The association was sustained after adjustments for age, weight, and BF%, reflecting an almost two-fold higher risk for low BMC given the presence of SO [OR = 1.92; 95% CI: (1.60–2.31), *p* < 0.05] (Table 2).

Finally, out of the total sample, 426 participants were affected by OSO, constituting 16.4%.

## 4. Discussion

The current study provided preliminary data with regard to the association between SO and reduced bone mass in treatment-seeking adults with obesity, and assessed the prevalence of a new phenotype, namely, OSO, in this population. Two main findings were revealed.

### 4.1. Findings and Concordance with Previous Studies

Firstly, our assessment revealed a prevalence of SO among treatment-seeking adults with obesity of nearly 35%, which is in line with what has been reported in several previous studies that used similar definitions that accounted for body mass (i.e., body weight or BMI) [17,19,31,32]. Generally speaking, a wide range of SO prevalence was reported in the literature and varies from 0% to 100% [18], depending on the SO definition used [18]. Higher prevalence is usually reported in studies that accounted for body mass (i.e., BMI); on the other hand, lower prevalence is reported in those that did not [18]. A low prevalence may also be explained by the use of definitions that have primarily been developed from studies on older cohorts, and these may not be applicable to younger adults [18].

Secondly, individuals with SO had a reduced mean BMC when compared to those without SO. Moreover, the SO group included more patients with low BMC (≈50%) with respect to their counterparts without SO (≈25%). In fact, the presence of SO seems to be strongly associated with the increase in the risk of having low bone mass. The underlying mechanism behind this crosstalk is still unclear; however, it seems that there is a bi-directional interaction between obesity, chronic inflammation, low bone mass, and sarcopenia [33,34,35], and we speculate that the coexistence of both obesity and sarcopenia under the so-called phenotype “SO” may have a synergistic effect, with chronic inflammation being a common “denominator” seen in both conditions, and may play an important role in bone remodeling, specifically in drive, exacerbating it toward a resorption state, leading to a reduction in bone mass [36]. Finally, regarding the new entity, the so-called OSO, that combines obesity, sarcopenia, and low bone mass, we found a prevalence of 16% in our total sample composed of treatment-seeking patients with obesity within a weight management setting. This finding is partially in line with a previous Korean study [37] which found a stronger association between SO and reduced bone mass (i.e., osteoporosis), but they found a lower prevalence of OSO of only 5% against ours of 16%. The reason behind the discrepancies between our study and other studies can be attributed to several factors. First and foremost, there was a difference in the studied samples in terms of ethnicity (i.e., Korean vs. Italian population) [37] and age groups (i.e., only included middle-aged and elderly) [37]. Moreover, we used different criteria for the definition of SO [38], as well as different indicators of bone mass (i.e., BMD) [37], meaning that these findings should be interpreted with caution, especially those related to the prevalence of OSO, since we are dealing with a new phenotype with no clear definition available to date [39]. In fact, defining OSO through a combined construct of two different dimensions, SO and low bone mass (i.e., BMC), can be critiqued and is open to discussion [25].

### 4.2. Study Strengths and Limitations

Our study has certain strengths. Firstly, it is one of the very few studies to investigate the association between SO and low bone mass in adults of both genders in a nutritional setting for obesity management, and to assess the prevalence of OSO [37]. Secondly, body composition was measured using DXA, which is a gold standard technique for bone mass measurement and regarded also as a precise method for fat and lean mass assessment, especially in patients with obesity and SO [40,41]. Thirdly we determined the low bone mass according to BMC tertile categorization of our specific population (i.e., obesity) and not on the t-score standard deviation of BMD compared to that of healthy young adults [42]. However, our study also has certain limitations. First and foremost, in defining OSO, we relied on a constructed definition that combined SO and low bone mass; however, to date, no precise definition of this new entity is available. This is especially important to note when the use of the term OSO is still debatable, with some researchers still arguing about the existence of OSO and whether it can be considered a new, distinct phenotype with specific clinical characteristics, rather than a simple association between its subparts (obesity, sarcopenia, and low bone mass) [26]. On the other hand, many other investigators support the opposite, recommending considering OSO as a separate clinical entity [21,22,23,25]. Secondly, our study, like other several papers published previously on OSO, fell short in exploring the association between OSO and clinical outcome (i.e., cardio-metabolic diseases), which we believe to be an issue that needs more investigation, particularly in light of very recent preliminary literature which supports this interaction—namely, the existence between OSO and a higher risk of hypertension, especially in women [24]. Thirdly, no functional test was performed to measure muscle strength, which is considered to be another necessary component in addition to muscle mass in SO diagnosis. Fourthly, our data were collected in a single unit, requiring external validation across other populations. Finally, no objective assessment of lifestyle parameters (dietary intake and physical activity levels) was performed, and these are factors known to affect body composition. In addition, performing a biochemical blood assessment for markers of chronic inflammatory status would allow us to investigate their central role in this new phenotype (i.e., OSO).

### 4.3. Potential Clinical Implications in Practice and New Directions for Future Research

The clinical implications of our findings are as follows: firstly, to highlight the importance of screening for SO in individuals affected by obesity and to also assess bone status in those individuals, since this condition (i.e., SO) seems to be strongly associated with impaired bone status; secondly, to start raising awareness among clinicians and patients regarding the presence of OSO in treatment-seeking adults with obesity. However, before confirming the clinical significance our findings, as a step toward translation into everyday clinical practice and to make firm and final recommendations in terms of diagnosis and management of OSO, relevant efforts should be put toward future research [43]. In particular, confirmation of the existence of OSO as a stand-alone, novel clinical entity is needed, followed by establishing screening tools and a clear definition of this new phenotype which could be easily used by healthcare professionals (e.g., clinicians, nutritionist, obesity specialists, orthopedists, rheumatologists, physiotherapists, etc.) to identify patients with obesity who are at higher risk of osteosarcopenia. Moreover, research which will assist in a better understanding of the underling mechanisms behind OSO and which can explain the interaction between muscle and bone under the umbrella of obesity is mandatory [44].

## 5. Conclusions

Osteosarcopenic obesity (OSO) is a new entity that is gaining interest in clinical settings that combines an excessive fat accumulation (i.e., obesity), reduced muscle mass and strength (i.e., sarcopenia), and low bone mass (i.e., osteopenia/osteoporosis) [45,46]. In our study, we reported a prevalence of this phenotype of 16% among adult treatment-seeking patients with obesity within a weight management nutritional setting. In addition, under this phenotype, we demonstrated that there is a strong association between sarcopenia and low bone mass. For this reason, our findings emphasize that complete body composition measurements (i.e., body fat, muscle, and bone) should be routinely performed in nutritional settings for weight management in patients affected by obesity.

## Figures and Tables

**Figure 1 life-14-00021-f001:**
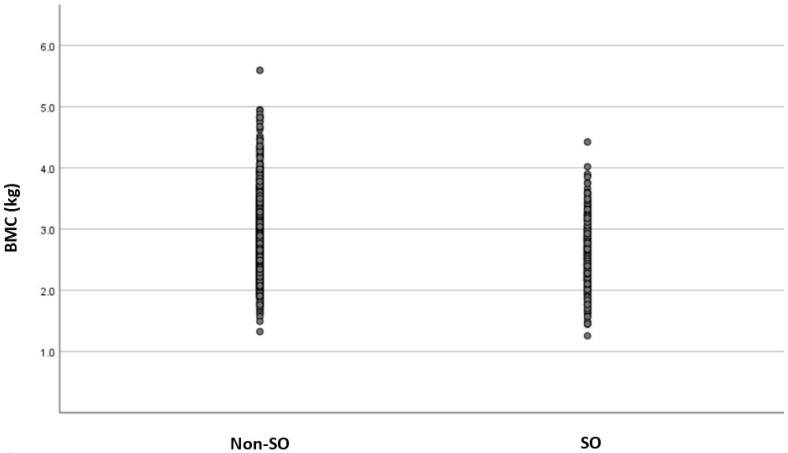
Scatter plot of mean BMC (kg) in groups with and without SO. SO = with sarcopenic obesity; Non-SO = without sarcopenic obesity; BMC = bone mineral content. Legend: the mean BMC expressed in kg was found to be significantly lower in the SO group when compared to the non-SO group.

**Figure 2 life-14-00021-f002:**
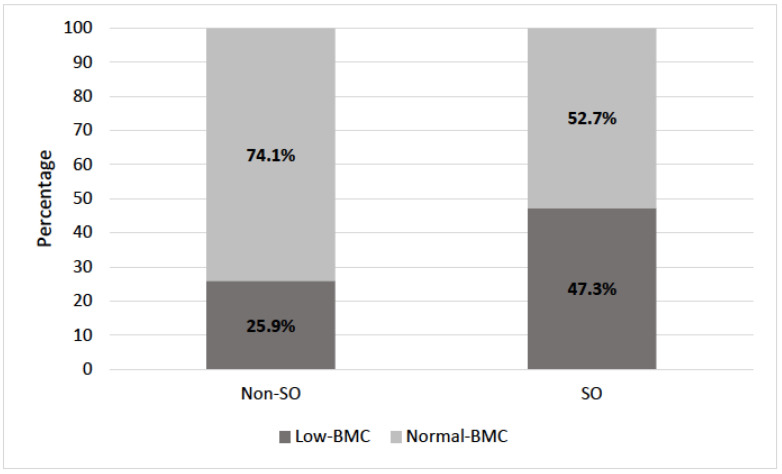
Prevalence of low BMC by category of SO. SO = with sarcopenic obesity; Non-SO = without sarcopenic obesity; BMC = bone mineral content. Legend: the SO group includes a higher number of individuals with low BMC when compared to those in the non-SO group.

**Table 1 life-14-00021-t001:** Demographics and anthropometrics of body composition characteristics of the study participants (*n* = 2604).

Variable	Total (*n* = 2604)	Non-SO (*n* = 1703)	SO (*n* = 901)	Significance *
Age	48.44 (14.31)	45.77 (13.53)	53.48 (14.38)	*p* < 0.01
Sex				X^2^ = 18.21; *p* < 0.01
Males	1112 (42.7)	676 (39.7)	436 (48.4)	
Females	1492 (57.3)	1027 (60.3)	465 (51.6)	
Weight (kg)	93.03 (13.23)	94.41 (12.8)	90.41 (13.59)	*p* < 0.01
Height (m)	1.65 (0.10)	1.68 (0.09)	1.61 (0.10)	*p* < 0.01
BMI (kg/m^2^)	33.87 (2.73)	33.53 (2.62)	34.52 (2.82)	*p* < 0.01
BF (kg)	39.57 (7.42)	39.02 (7.37)	40.61 (7.40)	*p* < 0.01
BF (%)	42.87 (7.31)	41.65 (7.31)	45.18 (6.72)	*p* < 0.01
LBM (kg)	50.71 (11.44)	52.54 (11.48)	47.25 (10.53)	*p* < 0.01
LBM (%)	54.16 (7.12)	55.32 (7.10)	51.98 (6.64)	*p* < 0.01
BMC (kg)	2.75 (0.56)	2.85 (0.57)	2.56 (0.46)	*p* < 0.01
BMC/h (kg/m)	1.65 (0.26)	1.69 (0.27)	1.58 (0.23)	*p* < 0.01
				X^2^ = 121.34; *p* < 0.01
Low BMC	867 (33.3)	441 (25.9)	426 (47.3)	
Normal BMC	1737 (66.7)	1262 (74.1)	475 (52.7)	

BMI = body mass index; BF = body fat; LBM = lean body mass; BMC/h = bone mineral content to height ratio. * The significance values refer to the tests of comparison between non-SO and SO.

**Table 2 life-14-00021-t002:** Logistic regression analysis showing the odds of low BMC in the presence of SO (*n* = 2604).

	Simple Model		Adjusted Model	
Variables	OR	95%CI	OR	95%CI
Age (years)	1.05	1.04–1.06	1.04	1.03–1.05
Weight (kg)	0.97	0.97–0.98	0.98	0.98–0.99
BF (%)	1.02	1.01–1.03	0.99	0.98–1.01
SO				
Non-SO	1.00		1	
SO	2.57	2.17–3.04	1.92	1.60–2.31

BF% = body fat percentage; SO = sarcopenic obesity. Legend: after adjustments for age, weight and BF%, the presence of SO increases the risk nearly two-fold of low BMC.

## Data Availability

The dataset in the present study is available upon request.

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
