# Peer review of "Association between Sarcopenia and Reduced Bone Mass: Is Osteosarcopenic Obesity a New Phenotype to Consider in Weight Management Settings?"

_life, 2023, doi:10.3390/life14010021_

Round 1

Reviewer 1 Report

Comments and Suggestions for Authors

In this research, De Lorenzo et al. showed that in adults with obesity, there is a greater prevalence of low bone mass in those with sarcopenic obesity than those without sarcopenic obesity. The authors suggest that this new phenotype called osteosarcopenic obesity should be defined and recognised in clinical settings. This research is of interest and relevance in the field since it recognises a highly prevalent condition in those with obesity which is found in 16% of the cohort studied. 

The manuscript is well written and has no major faults in the methodology to the best of my knowledge. My comments/suggestions for improvement:

I recommend that the authors state more clearly in their methods section how sarcopenic obesity was defined and categorised in the participants. 

Some improvements to grammar are: in line 85, which would be better if said: ‘patients were required to have a DXA scan’. Line 91, it is not clear if 2604 is the number of participants included or if these are the numbers considered for the study, please clarify.

Line 138 should say characteristics instead of characterizes.

For Table 1, indicate that the significance values refer to the test between SO and Non-SO. 

Overall, this manuscript addresses the issues of osteosarcopenic obesity which is an understudied condition that is shown to be common in those with obesity. This study provides a novel and important contribution to the field.  

Comments on the Quality of English Language

Some minor grammatical correction should be made which are pointed out in the comments. 

Author Response

In this research, De Lorenzo et al. showed that in adults with obesity, there is a greater prevalence of low bone mass in those with sarcopenic obesity than those without sarcopenic obesity. The authors suggest that this new phenotype called osteosarcopenic obesity should be defined and recognised in clinical settings. This research is of interest and relevance in the field since it recognises a highly prevalent condition in those with obesity, which is found in 16% of the cohort studied.

The manuscript is well written and has no major faults in the methodology to the best of my knowledge. My comments/suggestions for improvement:

Response: we thank the reviewer for the thoughtful comments that we did our best to address.

I recommend that the authors state more clearly in their methods section how sarcopenic obesity was defined and categorised in the participants.

Response: We added the requested information in the Method section (Page 3, paragraph 4).

Some improvements to grammar are: in line 85, which would be better if said: ‘patients were required to have a DXA scan’.

Response: we replaced the statement as suggested (Page 2, paragraph 5).

Line 91, it is not clear if 2604 is the number of participants included or if these are the numbers considered for the study, please clarify.

Response: we reported that 2,604 participants representing both genders were included as the study sample (Page 3, paragraph 1).

Line 138 should say characteristics instead of characterizes.

Response: we corrected as indicated (Page 4, Table 1).

For Table 1, indicate that the significance values refer to the test between SO and Non-SO.

Response: we specified that the significance values refer to the test between Non- SO and SO (Page 4, Table 1).

Overall, this manuscript addresses the issues of osteosarcopenic obesity which is an understudied condition that is shown to be common in those with obesity. This study provides a novel and important contribution to the field.

Response: we thank the reviewer for the appreciation.

Reviewer 2 Report

Comments and Suggestions for Authors

This is an retrospective study aiming to explore the interrelationship between obesity, sarcopenia and reduced bone density. 

Few comments: 

1) The authors discuss the term Osteosarcopenic obesity, however, it is unclear whether this represents a distinct and separate clinical entity with different clinical characteristics and outcomes rather than a radiological  combination of its subparts. In this regard, the manuscript falls short in exploring associations with lifestlyle, clinical, and biochemical parameters (acknowledged as a limitation) to better describe OSO as a clinical diagnosis. In this regard, the authors need to expand their discussiion along this lines highlighting the debate of whether (and why) OSO is clinically different. 

2) Can the authors correlate with clinical outcomes (such as prevalance of cardiometabolic diseases for each group)?

3) The clinical significance and translational potential to everyday clinical practice alongside clinical recommendations for the diagnosis and/or management of OSO as a condition need to be discussed. 

4) Quality of figures require graphical improvement. Please provide more information in the figure and table legends. Ideally these should be stand alone items with no need for reference to the main text. 

5) Overall, it is unclear what this study brings to the literature. 

Author Response

This is a retrospective study aiming to explore the interrelationship between obesity, sarcopenia and reduced bone density.

Few comments:

1) The authors discuss the term Osteosarcopenic obesity, however, it is unclear whether this represents a distinct and separate clinical entity with different clinical characteristics and outcomes rather than a radiological combination of its subparts. In this regard, the manuscript falls short in exploring associations with lifestyle, clinical, and biochemical parameters (acknowledged as a limitation) to better describe OSO as a clinical diagnosis. In this regard, the authors need to expand their discussion along this lines highlighting the debate of whether (and why) OSO is clinically different.

Response: we thank the reviewer for the thoughtful insight. This point now is raised and extensively discussed in the discussing section (Page 7, paragraph 2). 

2) Can the authors correlate with clinical outcomes (such as prevalence of cardio metabolic diseases for each group)?

Response: due to the design and the retrospective nature of our study we are not in the position to correlate OSO with clinical outcome such as cardio-metabolic diseases. Now we have clearly mentioned it among the limitations (second) (Page 7, paragraph 2).

3) The clinical significance and translational potential to everyday clinical practice alongside clinical recommendations for the diagnosis and/or management of OSO as a condition need to be discussed.

Response: This comment has been discussed as suggested in the context of the subsection 4.3. (Page 7, paragraph 3).

4) Quality of figures requires graphical improvement. Please provide more information in the figure and table legends. Ideally these should be stand-alone items with no need for reference to the main text.

Response: The quality of the figures has been improved and legends for figures 1 and 2 and table 2 are now added.

5) Overall, it is unclear what this study brings to the literature.

Response: we highlighted in the section 4.3. what this study brings to literature in terms of clinical practice as well as the needed of future research on the topic (Page 7, paragraph 3).

Round 2

Reviewer 2 Report

Comments and Suggestions for Authors

Thanks for addressing the issues raised.